# One Hundred Years of Progress and Pitfalls: Maximising Heterosis through Increasing Multi-Locus Nuclear Heterozygosity

**DOI:** 10.3390/biology13100817

**Published:** 2024-10-12

**Authors:** Brendan F. Hallahan

**Affiliations:** Public Analyst’s Laboratory, St. Finbarr’s Hospital, Cork T12 XH60, Ireland; brendan.hallahan@hse.ie

**Keywords:** hybrid vigour, polyploidy, heterozygosity, allele number, progressive heterosis

## Abstract

**Simple Summary:**

Humans are *diploid* organisms, carrying one set of chromosomes from our mother and one from our father. Plants can exist as *diploid* or *polyploid*, carrying multiple chromosome copies. Plant breeders can cross *polyploid* crops in unique and beneficial ways that are not possible in *diploid* crops. It will be necessary to produce food in a more sustainable manner, with less virgin land turned over and fewer resources consumed, therefore the study of hybrid *polyploid* crops is essential for the improvement of our food system.

**Abstract:**

The improvement in quantitative traits (e.g., yield, size) in F_1_ offspring over parent lines is described as hybrid vigour, or heterosis. There exists a fascinating relationship between parental genetic distance and genome dosage (polyploidy), and heterosis effects. The contribution of nuclear heterozygosity to heterosis is not uniform across diploid and polyploid crops, even within same species, thus demonstrating that polyploid crops should be part of any discussion on the mechanisms of heterosis. This review examines the records of correlating heterosis with parental genetic distance and the influence of adding supplementary genomes in wide crosses. Increasing nuclear heterozygosity through parental genetic distance has been shown to be an imperfect predictor for heterosis in a variety of commercial crops such as maize, rice, and pepper. However, increasing the ploidy level raises the maximum number of alleles that can be harboured at any one locus, and studies on crops such as oilseed rape, potato, alfalfa, maize, and rice have demonstrated that heterosis may be maximised upon increasing multi-locus nuclear heterozygosity. The novel heterotic phenotypes observed above the diploid level will contribute to our understanding on the mechanisms of heterosis and aid plant breeders in achieving the righteous goal of producing more food with fewer inputs.

## 1. Introduction

In 1912, Edward M. East and Herbert K. Hayes of the United States published the first comprehensive review of hybrid breeding in crops of commercial interest, chiefly maize and tobacco [1]. The phenomenon of hybrid vigour, or heterosis, was of great interest to “add many millions of dollars annually to the nation’s resources”. It was well established by then that a so-called state of heterozygosis in a first-generation offspring (filial 1; F_1_) from inbred parents would regularly produce taller and/or higher yielding crops. East and Hayes interrogated the data as they knew this, to try and determine why certain “subdivisions” of species produced vigorous offspring upon cross-fertilisation, while others produced weaker crops or were completely unviable. Over 100 years later, there have been considerable developments on the genetic and epigenetic consequences, as well as the molecular mechanisms at play when plants are cross-fertilised.

This review brings together the published records of wide crosses at various ploidy levels where there were no fatal fertility problems. The review will only consider crosses through sexual hybridisation, not regenerated plants from artificial means such as somatic hybridisation or embryo sac chromosome doubling [2]. This review highlights how limits for heterosis through increasing multi-locus nuclear heterozygosity at the diploid level may be overcome through whole genome duplication events or the addition of supplementary genomes. This review begins with a framing of the heterosis phenomenon with a number of valuable references if the reader wishes to learn more, but does not set out to be an all-encompassing review of the potential mechanisms at play; for that the author suggests the excellent reviews by Schnable and Springer [3] and Labroo et al. [4]. Discussions regarding different techniques for estimating genetic distance and their usefulness in predicting heterosis have been covered by dos Santos Dias et al. [5]. Finally, the author wishes to complement previous work on this topic by Washburn and Birchler [6], which was the inspiration for this review.

## 2. Crop Improvement through Heterosis

An F_1_ hybrid offspring is the genetic composite of the biparental nuclear genomes inherited from both parents and the maternal cytoplasmic genome. An F_1_ hybrid displays heterosis when it displays enhanced characteristics over one, or both, parents [7]. From a commercial viewpoint, a more practical definition of heterosis is the superior performance of an F_1_ hybrid over the best parent [8]. Commercial F_1_ hybrid seed has contributed between 10–50% of yield increases since the mid-20th century for many crops, including bread wheat (*Triticum aestivum*), maize (*Zea mays*), sorghum (*Sorghum bicolor*), soybean (*Glycine max*), sugar beet (*Beta vulgaris*), and upland cotton (*Gossypium hirsutum*), to name just six [9,10,11,12,13,14,15]. While the significance of heterosis to commercial crops is widely documented, an all-inclusive mechanism explaining heterosis is lacking. See Box 1 for a description of the molecular models used to explain the heterosis phenomenon and valuable references where these models have been tested and discussed. Whatever the contributing molecular event(s), this will be followed by a functional mechanism that will give rise to heterosis. Proposed mechanisms include the following: optimisation of rate-limiting enzyme activity in maize [16]; increase in gibberellic acid in maize [17] and wheat [18]; epigenetic changes to circadian regulatory genes in tetraploid crosses between *Arabidopsis thaliana* and *Arabidopsis arenosa* [19]; specific leaf transcriptomics in maize and *Arabidopsis thaliana* [20]; improved energy-use efficiency for growth and protein metabolism [21]; changes in 24 nt small-interfering RNA (siRNAs) in maize [22]; repression of stress-responsive genes in *Arabidopsis thaliana* [23]; and an increase in potassium uptake and transport in tobacco (*Nicotiana tabacum*) [24]. From this list, it is clear that there exists specific heterotic traits that arise from specific genetic or epigenetic changes in specific F_1_ hybrid plants.

Box 1Molecular models to explain heterosis.
**Contribution of dominance to heterosis**
The dominance theory of heterosis may be most 
easily comprehended as the opposite of inbreeding depression. It suggests 
that hybridising genetically distinct genomes will lead to heterosis. 
Assuming both parents harbour different recessive, slightly deleterious 
alleles in a homozygous state, these will be complemented by a dominant, 
superior copy from the other parent in the F_1_ hybrid. Thus, the 
deleterious effect of these recessive alleles will be masked in the 
heterozygous state. The practical challenge of achieving a homozygous (or 
near-homozygous) condition in the parent lines in order to demonstrate a 
heterosis effect on crop yield was clearly demonstrated in the pioneering 
work of George H. Shull presented to the American Breeder’s Association [25,26].
**Contribution of over-dominance to heterosis**
It was addressed in the early literature by 
Shull [27], Jones [28], and East [29] that a simple dominance mode of action does 
not account for all the heterosis effects observed in many F_1_ 
plants. The over-dominance theory proposes that heterozygosity at a specific 
locus or loci will lead to heterosis, while either homozygous condition will 
not. This was skilfully demonstrated in tomato (*Solanum lycopersicum*) 
by Krieger et al. [30], where considerable yield heterosis was 
demonstrated when there was heterozygosity for loss-of-function alleles of *SINGLE 
FLOWER TRUSS *(*SFT*). F_1_ hybrids homozygous for the *sft* 
mutation do not display heterosis. Linked genes can make the task of 
determining dominant from over-dominant mechanisms difficult. For example, it 
was reported that over-dominant effects at several quantitative trait loci 
(QTL) were responsible for maize heterosis [31]. These may be dominant effects. The QTL with 
the largest influence identified were later found to be an effect called 
“pseudo-overdominance”: the QTL were further dissected into (at least) two 
distinct genes that were in a state of repulsion phase linkage. These linked 
loci both contribute to heterosis via dominant mechanisms [3,32].
**Contribution of epistasis to heterosis**
Further nuance is required when considering 
the epistasis theory of heterosis. This theory is defined as novel 
interactions between alleles at two or more different loci leading to a 
heterotic effect. For example, an over-dominant acting locus may interact 
with a dominant-acting locus elsewhere in the genome of an F_1_ 
hybrid, thus contributing to heterosis [33]. This theory was first described in detail by 
Powers [34]. Epistasis mechanisms have been proposed to account for 
the majority of the heterotic effects in certain crosses of wild tomato *Lycopersicon 
hirsutum *through the use of near-isogenic lines [35] and in 
rice through chromosome segment substitution lines [36]. Both of 
these approaches are examples of marker assisted selection, which is 
admirable due to the large number of controlled crosses required to isolate 
specific genomic regions and their associated allelic frequency.
**Contribution of epigenetics to heterosis**
While genetically disparate individuals may 
induce heterosis through genetic mechanisms, it is probable that increased genetic differences are also associated with increased epigenetic differences [37]. In the context of heterosis, epigenetic effects can be 
defined as heritable changes in gene activity which are unrelated to 
underlying changes in DNA sequences [38]. To determine if parental epigenetic effects 
alone can cause heterosis, it is necessary to maintain a population of inbred 
lines that vary only for a segregating epigenetic marker. Such epigenetic 
recombinant inbred lines (epiRILs) in *A. thaliana* have been crossed 
with the relevant wild-type accession to demonstrate how changes in DNA 
methylation, independent of genetic differences, can positively influence 
plant height [39] and leaf area [40,41]. Attempts to create epiRILs in agronomically 
important crops like maize [42] and rice [43,44] have been met with obstacles such as strong 
lethality following alterations to the DNA methylation network.

## 3. Polyploid Heterosis

Increasing the number of chromosome sets in the nucleus, including whether the extra genome(s) is inherited maternally or paternally, can provide a neat experimental design to further test the limits of heterosis. In *Arabidopsis thaliana*, for example, there exists a parent-of-origin genome dosage effect on F_1_ triploid heterosis. If an F_1_ triploid inherits two paternal genomes, the heterosis effect on seed size and leaf area exceeds that seen in F_1_ diploid hybrids. However, this heterosis effect is ablated or indeed reversed if the F_1_ triploid hybrid inherits two maternal genomes [45,46,47]. It is hypothesised that the ratio of maternal:paternal genomes in the endosperm has a pronounced impact on inter-ploidy F_1_ seed size, with larger seeds giving rise to larger plants, although growth rates remain the same, demonstrating a “hidden” heterosis trait at the diploid level. The functional mechanism behind this effect on seed size remains to be fully explained [47]. A similar genome dosage effect on F_1_ triploid heterosis has been demonstrated in maize, although some heterotic traits are genotype dependent [48]. 

At the tetraploid level, it has been demonstrated in several plants (discussed below) that increasing heterozygosity in a double-cross (beyond a single cross) can increase the heterosis effect. This phenomenon has been termed ‘progressive heterosis’ and is accepted to be absent in diploids. This is because a tetraploid double-cross F_1_ hybrid can harbour up to four different alleles at any one locus (Figure 1). This suggests that increasing multi-locus nuclear heterozygosity by adding extra genomes could maximise heterosis.

## 4. Increasing Multi-Locus Nuclear Heterozygosity and Heterosis

The F_1_ progeny of genetically disparate parents will harbour considerable nuclear heterozygosity. According to the dominance theory of heterosis, any increase beyond the homozygous state at a locus will maximise the heterosis effect. Thus, heterosis should correlate with nuclear heterozygosity. This experiment has been run a number of times with a diverse range of plant species—those both autogamous and allogamous in nature, with different growing habits, and at a range of ploidy levels. 

For diploid plants of the same species (“autodiploid”), crossing genetically diverse parent lines in a single cross to significantly increase nuclear heterozygosity is an imprecise method of maximising heterosis. Crossing genetically diverse parent lines in maize appears to maximise heterosis, although a certain threshold is reached, according to Moll [49]. Early work with restriction fragment link polymorphisms (RFLPs) also indicated a significant [50] and intermediate [51] relationship between genetic distance and heterosis. However, subsequent work with a variety of molecular markers has reached the opposite conclusion [52,53,54,55,56]. Perhaps the most comprehensive work was by Reif et al. [55], where maize F_1_ hybrids of European and United States inbred lines were examined at multi-location field trials. Likewise, for pepper (*Capsicum annuum*) [57], rice (*Oryza sativa*) [58], soybean (*Glycine max*) [59], sesame (*Sesamum indicum*) [60], white clover (*Trifolium repens*) [61], chickpea (*Cicer arietinum*) [62], eggplant (*Solanum melongena*) [63], pearl millet (*Pennisetum glaucum*) [64], broad bean (*Vicia faba*) [65]), oil palm (*Elais guineensis*) [66], pigeon pea (*Cajanus cajan*) [67], melon (*Cucumis melo*) [68], the amphidiploid Ethiopian mustard (*Brassica carinata*) [69], and the model organism *Arabidopsis thaliana* [70,71], heterosis at the diploid level is not maximised through crossing genetically diverse parent lines. Outliers to this broad conclusion are found in diploid cocoa (*Theobroma cacao*) [72,73] and diploid Robusta coffee (*Coffea canephora*) [74], as well as limited evidence in sorghum (*Sorghum bicolor*) [75]. Likewise, in diploid sunflower (*Helianthus annus*) there were early reports that heterosis correlates with genetic distance, although the authors found differentiating highly heterozygous open-pollinated varieties difficult [76]. Subsequent work with elite Russian varieties also found a significant correlation [77]. Interesting results have been reported in diploid carrot (*Daucus carota*) where RFLP markers suggest a relationship between genetic distance and total yield in F_1_ hybrids, but amplified fragment length polymorphism (AFLP) markers—which are generally accepted to be more sensitive [78]—do not [79]. Interspecific hybridisation (mating across taxonomically-defined species boundaries) may produce viable plants. At the diploid level (“allodiploid”) there are some prominent examples of heterosis when crossing members of the radish (*Raphanus*) and *Brassica* genus [80,81]. However, pre- and post-fertilisation barriers commonly prevent successful allodiploid formation. 

There is a strong record of maximising heterosis through crossing genetically diverse parent lines in polyploid plants (Table 1). The F_1_ hybrids in Table 1 represent controlled crosses among well-defined genotypes, i.e., inbred lines, near-inbred lines, or elite commercial varieties, not distant crosses with wild relatives. Progressive heterosis has been recorded in many crops which are commercially grown at the tetraploid level, such as potato [82] and alfalfa [83,84]. Of note, the effect has also been recorded in maize [53,85] and rice [86,87], which gives further credence to the view that heterosis at the diploid level does not predict heterosis at the polyploid level. It is important to highlight when authors investigate the same plant species but use different methodologies to determine genetic differences. In oilseed rape (*Brassica napus*), contrasting conclusions have been drawn by Diers et al. [88] and Riaz et al. [89], with the former using RFLPs and the latter using sequence-related amplified polymorphism (SRAP) markers. Diers and colleagues conclude that there is no meaningful relationship between genetic distance and heterosis, whereas Riaz and colleagues found a significant relationship between the two. An earlier study using knowledge of the geographical origin of oilseed rape varieties also found a significant relationship [90]. For bread wheat using either PCR sequence tagged sites (STSs) or RFLPs, both demonstrate an absence of any relationship between genetic distance and heterosis (Table 1). 

Why do many polyploid crops tend to demonstrate heterosis upon large increases in nuclear heterozygosity, but diploid crops do not? Taking the examples in Table 1, it appears that for many crops, increasing the number of different alleles has a positive dosage effect on vigour. Intralocus (i.e., epistatic) interactions have been attributed to the heterosis effects in polyploid potato [91], although it has been demonstrated that a three-way cross may be no more beneficial than a two-way cross (Figure 1) [92], but this finding appears to be in the minority (Table 1). This suggests that favourable alleles are scattered across the genome and their positive impact are fully actualised in a polyploid plant. However, concluding that multiple alleles at a certain locus are the chief driver of heterotic effects may overlook the presence of a tight collection of linked genes along a small chromosomal segment—a ‘linkat’—which may be indistinguishable from multiple alleles [93]. In alfalfa, it has been suggested that the number of different alleles at a locus is only one part of the explanation for progressive heterosis, and the effect of complementary action between genes could be more influential [94]. For example, suppose heterosis arising from a certain gene action is present in equivalent diploid and autotetraploid plants, but the differences between disomic and tetrasomic segregation will produce a higher frequency of heterozygotes in the autotetraploid; thus, an F_1_ arising from tetrasomic segregation is more likely to possess loci containing at a least one dominant allele, potentially masking inferior, recessive alleles (i.e., over-dominance) [95] (see schematic below). In addition, the influence of complementary gene action on heterosis in alfalfa is clearly outlined by Bingham et al. [94] using the example of “two-allele populations”. Taking equivalent diploid and autotetraploid populations which possess a maximum of two different alleles at a locus (e.g., genotype Aa and genotype AAaa), populations are maintained for a number of generations (e.g., self and sib-mating). Bingham and colleagues discuss the expression of heterosis vs. inbreeding depression in these populations and explain that the presence of different alleles in the population (A, A_1_, A_2_, A_3_, …A_j_) existing at ‘linkats’, which display additive effects on the phenotype, are likely responsible (an additive effect on the phenotype would manifest as a linear increase, whereas a non-additive effect on the phenotype would be non-linear). The tetrasomic segregation of favourable alleles at ‘linkats’ in autotetraploids present greater opportunities for complementary gene actions than seen in diploids. The converse of this, upon repeated selfing, autotetraploid alfalfa display inbreeding depression at much higher levels than would be expected considering they take a longer time to reach homozygosity than an equivalent diploid [96]. The difference in levels of inbreeding depression between identical genotypes cannot be attributed to recessive alleles alone, but rather the loss of complementary gene action in the autotetraploid as allelic dosage changes faster than homozygosity [94]. Remarkably, a swift display of inbreeding depression has also been reported in autotetraploids of maize [97] and sugar beet [98], while autotetraploids of rye (*Secale cereale*) show a similar level of inbreeding depression to their equivalent diploids [99,100]. This suggests the breakdown of complementary gene action upon inbreeding is not a rare occurrence in autotetraploids.

If autotetraploidy is a reliable route to maximising heterosis, what are the barriers to widespread adoption of double-cross autotetraploid hybrids? The answer can be species or genotype specific. While potato, alfalfa, and perennial ryegrass are commercially grown at the tetraploid level, maize is commercially grown at the diploid level. In maize, increasing the ploidy level can have an immediate, negative impact on plant growth in some genotypes [101]. Tetraploid maize has lower fertility than equivalent diploids [102] and they can potentially exhibit ‘double reduction’ in meiosis, where incorrect separation of sister chromatids during the first meiotic division means the resulting gamete contains both sister chromatids [103]. Thus, the progeny from a double-cross tetraploid maize could possess different genotypes. Doubling the genome dosage can likewise have a detrimental effect on various morphological characteristics in some plants [104]. Autotetraploid rice has a less than ideal fertility level for commercial purposes [105]. Upon increasing the genome dosage, a tetraploid plant typically displays a slower growth rate [106]. This change would need to be incorporated into agronomic practices, e.g., growing degree days and chemical applications. Lastly, selecting inbred lines in a heterozygous autotetraploid population can be challenging in certain circumstances [107]. If the objective is to remove a deleterious allele at a certain locus (i.e., achieve complete dominance), tetrasomic segregation in an autotetraploid makes selection more cumbersome than disomic segregation in an equivalent diploid. For example, assume the hypothetical allele *A* shows incomplete dominance over allele *a*. Upon selfing, segregation at a heterozygous locus *Aa* in a diploid will segregate so that ^1^/_4_ of the progeny will show complete dominance.
♀Aa♂AAAAaaAaaa

The same heterozygous locus *AAaa* in an autotetraploid will produce far more undesirable genotypes harbouring the recessive allele upon selfing (assuming full tetrasomic segregation), and only ^1^/_36_ of the progeny will show complete dominance.
♀AAAaAaAaAaaa♂AAAAAAAAAaAAAaAAAaAAAaAAaaAaAAAaAAaaAAaaAAaaAAaaAaaaAaAAAaAAaaAAaaAAaaAAaaAaaaAaAAAaAAaaAAaaAAaaAAaaAaaaAaAAAaAAaaAAaaAAaaAAaaAaaaaaAAaaAaaaAaaaAaaaAaaaaaaa

Wide crosses within cultivated wheat genotypes do not reliably induce heterosis (Table 1). Raising the “yield plateau” in wheat [108,109] to meet future demand is limited by the narrow genetic base among cultivated genotypes from a loss in variation during domestication and selection [110,111,112,113]. This suggests wheat breeders may need to focus on organising present germplasm into heterotic groups [114] or look to wild relatives to introduce novel alleles [115,116,117,118]. However, a commendable research effort producing over 1500 wheat F_1_ hybrids from European genotypes demonstrated the potential for an approximate 10% increase in yield above average, and the majority of heterotic effects were attributed to epistatic interactions [119]. This shows the complex heritability of yield in wheat, likely controlled by multiple loci, each with small effects. A hybrid between wheat and rye, *Triticale*, is common across Europe as a feed, energy, and food crop [120]. Also present at the allohexaploid level, commercial *Triticale* genotypes can exhibit heterosis in the F_1_ following crosses with genetically diverse parents [121]. The link was small but significant, suggesting that seeking out genetic diversity among cultivated *Triticale* could be an essential prerequisite when breeding for improved varieties. Likewise, cultivated forage grass Timothy (*Phleum pratense*) is an allohexaploid crop, and starting with parents with commercially favourable yield it is possible to improve yield further upon wide crosses [122]. This suggests that despite the presence of homeologous genomes and after a certain amount of selection, these crops continue to harbour potentially recessive allelic combinations.

**Table 1 biology-13-00817-t001:** An overview of the relationship between nuclear heterozygosity and heterosis in polyploid crops.

Ploidy Status	Crop	Association Observed between Crossing Genetically Diverse Parent Lines in a Single Cross and Heterosis in F_1_?	Reference
Autotriploid	Sugar beet (*Beta vulgaris* subsp. *vulgaris*)	No	[123]
Autotetraploid	Rye (*Secale cereale*)	Yes	[99]
	Potato (*Solanum tuberosum*)	Yes	[82,92,124]
		No	[125]
	Alfalfa (*Medicago sativa*)	Yes	[83,84]
	Maize (*Zea mays*)	Yes	[53,85]
	Perennial ryegrass (*Lolium perenne*)	Yes	[126]
	Rice (*Oryza sativa*)	Yes	[86,87]
	Bahiagrass (*Paspalum notatum*)	Yes	[127]
Allotriploid	Willow ((*Salix koriyanagi* × *S. purpurea*) × *S. miyabeana*	Yes	[128]
Allotetraploid	Peanut (*Arachis hypogaea*)	No	[129]
	Oilseed rape (*Brassica napus*)	Yes	[89,90]
		No	[88]
	Arabica coffee (*Coffea arabica*)	Yes	[130]
	Upland cotton (*Gossypium hirsutum*)	No	[131]
Allohexaploid	Bread wheat (*Triticum aestivum* subsp. *aestivum*)	No	[132,133]
	*Triticale*	Yes	[121]
	Timothy (*Phleum pratense*)	Yes	[122]

## 5. Commercial Breeding

For a hybrid breeding program to be commercially successful, the F_1_ must possess economically valuable traits that cover the cost of the program in question. The attributes of the crop (e.g., outcrossing or self-fertilising, availability of cytoplasmic male sterile lines, annual or perennial) will determine the most appropriate breeding plan to adopt. Regardless, F_1_ hybrid breeding programs can be made more efficient with some form of predictive value. Determining the genetic relatedness between parent lines can help develop clearly defined populations or heterotic groups, from which superior F_1_ hybrids can been formed through knowledge of combining ability.

To support commercial breeding programs in this endeavour, there exists a number of molecular biology tools. For example, gene expression analysis of messenger RNA (mRNA) can reveal transcriptional variation between parents which may be exploited for a heterotic effect in the F_1_ [134,135]. Regardless of ploidy level, this approach is difficult as it requires a specific developmental stage to be sampled, on precise tissue, across similar environmental conditions. Alternatively, genomic DNA can be sampled to create molecular markers (e.g., RFLPs, SSRs) across a set of parent lines and offspring which can be correlated with heterosis. High-throughput sequencing technologies can improve the efficiency of this process, but random markers may demonstrate poor linkage to QTL controlling traits of interest as more genetic backgrounds are introduced [136]. An improved use of molecular markers includes genomic selection, where a computer model is trained on high-density markers throughout the entire genome in combination with phenotype scores [137]. As new genotypes are added to the model, accurate phenotype prediction can improve commercial breeding resources [138,139]. In addition, fluorescent in situ hybridisation (FISH) can determine physical changes in chromosomes and identify copy number differences between parent lines and progeny. Such labelled DNA is useful to help visualise the presence or absence of chromosomes or chromosomal fragments. However, this technique is open to added complexity in polyploid plants due to genes with multiple copies throughout the genome from chromosomal duplications [140]. Combining genetic and metabolite markers to predict heterosis (e.g., single nucleotide polymorphisms (SNPs) with sugar, amino acid, organic acid profiles) can create “biomarkers” [141,142,143], although knowledge of the essential metabolites in the crop of interest and when to sample for the varying analyses is challenging.

A simple predictor of heterosis could be genetic distance between parents, as wide crosses could exploit novel gene recombination. Following such a cross, nuclear heterozygosity is substantially increased. However, diploid crops cannot exploit this level of heterozygosity whereas many polyploid crops can (Table 1). For the majority of diploid crops, wide crosses potentially introduce multiple loci with negative or neutral effects on heterosis. The documented exceptions to this observation are all out-crossing diploids (cocoa, coffee, sunflower), suggesting a preference to maintain a highly heterozygous genotype for plant development. Thus, together with all the molecular biology tools available to streamline the development of valuable F_1_ hybrids, polyploid breeding programs have potentially more favourable crosses at their disposal than diploid breeding programs.

## 6. Conclusions

To explain the presence or absence of heterosis, a number of molecular theories have been tested (Box 1). We now know that studying diploid crops in isolation means a number of hidden heterotic effects are absent from the discussion [6]. East and Hayes [1] noted that the heterozygous condition was generally favourable, although some F_1_ hybrids from genetically dissimilar parents showed little or no increase in vigour. If East and Hayes could have converted these diploid maize lines to the tetraploid state, heterosis may have been observed in the F_1_ [53,85]. This is because polyploid crops, even commercial lines, may be improved further through increasing multi-locus nuclear heterozygosity. Over 100 years following East and Hayes’ influential publication, adding a genome dosage variable to the heterosis equation typifies this essential, albeit enigmatic, tool for plant breeders.

## Figures and Tables

**Figure 1 biology-13-00817-f001:**
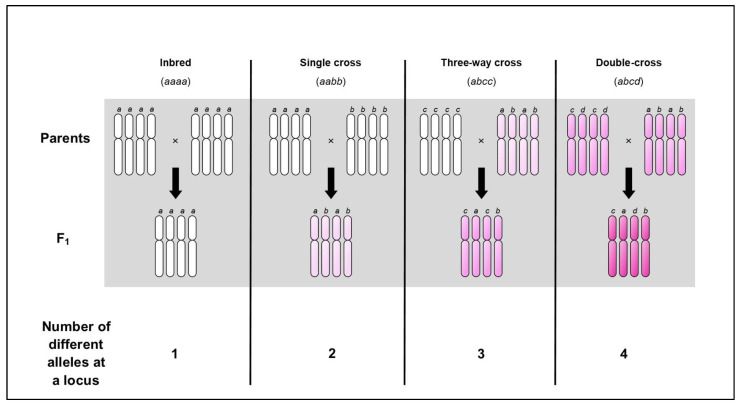
F_1_ autotetraploid from a double-cross can harbour four different alleles at a locus. An autotetraploid contains four identical chromosome sets within its nucleus. Let us propose a hypothetical gene at a certain locus that has four different alleles, denoted a, b, c, and d. Inbred (100% homozygous) lines will possess a mono-allelic locus for this gene, i.e., a simplex genotype meaning four identical copies of one allele (aaaa and bbbb). Following cross-fertilisation, the single cross F_1_ hybrid will possess a di-allelic locus for this gene, i.e., a duplex genotype meaning two copies of two different alleles (aabb). This single cross F_1_ hybrid can be crossed to a different inbred line (cccc) and the resulting three-way cross F_1_ hybrid will possess a tri-allelic locus for this gene, i.e., a trigenic genotype meaning one copy of two different alleles and two identical copies of one allele (abcc). Alternatively, the single cross F_1_ hybrid can be crossed to a different single cross F_1_ hybrid (ccdd) and the resulting double-cross F_1_ hybrid will possess a tetra-allelic locus, i.e., a tetragenic genotype meaning one copy of four different alleles (abcd). The increasing intensity of the colour pink reflects the increasing levels of heterozygosity.

## Data Availability

No new data were created or analysed in this Review. Data sharing is not applicable.

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
