# Peer review of "One Hundred Years of Progress and Pitfalls: Maximising Heterosis through Increasing Multi-Locus Nuclear Heterozygosity"

_biology, 2024, doi:10.3390/biology13100817_

Round 1

Reviewer 1 Report

Comments and Suggestions for Authors

The author discusses the concept of hybrid vigor, or heterosis, which refers to the improvement of quantitative traits in F1 offspring over parent lines and highlights the relationship between parental genetic distance, genome dosage, and heterosis effects. The author also explores the correlation of heterosis with parental genetic distance and how increasing ploidy level can maximize heterosis by increasing multi-locus nuclear heterozygosity. However, this draft still lacks information regarding three points, as below:

1.      Practical discussion regarding heterosis prediction in a crop species is important. Heterosis prediction is essential in a routine commercial seed business, as a large scale of hybridization followed by extensive field evaluation is time consuming and resource intensive. Thus, I suggest the author provide additional sections to discuss several molecular and biometric models to predict heterosis in either diploid or polyploid crops.

2.      Providing more graphical illustrations with detail captions, like Figure 1, will make the manuscript more understandable for general readers.

3.      I suggest authors provide an additional section to showcase the outlook of heterosis issue in both diploid and polyploid crops.

Author Response

The author discusses the concept of hybrid vigor, or heterosis, which refers to the improvement of quantitative traits in F1 offspring over parent lines and highlights the relationship between parental genetic distance, genome dosage, and heterosis effects. The author also explores the correlation of heterosis with parental genetic distance and how increasing ploidy level can maximize heterosis by increasing multi-locus nuclear heterozygosity. However, this draft still lacks information regarding three points, as below:

  1. Practical discussion regarding heterosis prediction in a crop species is important. Heterosis prediction is essential in a routine commercial seed business, as a large scale of hybridization followed by extensive field evaluation is time consuming and resource intensive. Thus, I suggest the author provide additional sections to discuss several molecular and biometric models to predict heterosis in either diploid or polyploid crops.

Response: Thank you for this suggestion. Section 5 Commercial Breeding line 281 has been changed to include a discussion on the various molecular methods used by breeders to help predict heterosis. This includes technologies such as transcriptomic analysis and genomic selection. This has greatly improved this section as it goes on to emphasise the central point of the manuscript: polyploid breeding programs can exploit much wider crosses than diploid breeding programs.

  1. Providing more graphical illustrations with detail captions, like Figure 1, will make the manuscript more understandable for general readers.

Response: Thank you for this suggestion and recognition of the detail added to Figure 1. An early version of this manuscript was drafted to include graphic demonstrations of the molecular theories explaining heterosis (dominance, over-dominance, and epistasis). However, this was dropped as such illustrations already exist across second- and third-level textbooks and multiple other scientific papers. Included in the new manuscript is an illustration of disomic inheritance in selfed diploids and tetrasomic inheritance in selfed autotetraploids in section 4 Increasing multi-locus nuclear heterozygosity and heterosis line 230.

  1. I suggest authors provide an additional section to showcase the outlook of heterosis issue in both diploid and polyploid crops.

Response: Thank you for this suggestion. The closing paragraph of section 5 Commercial Breeding line 304 in the revised manuscript addresses this point in regard to all potentially favourable crosses available in diploid and polyploid breeding programs.

Reviewer 2 Report

Comments and Suggestions for Authors

I have thoroughly reviewed your article and would like to commend you on the comprehensive nature of your work. The writing is clear, and the practical aspects are well-explained.

However, I have a few questions regarding hybrid seeds and heterosis in tetraploids and polyploids that I believe could enhance the depth of your manuscript. Specifically:

  1. How does the production of inbred lines differ between diploids and tetraploids? Are there any significant distinctions in this process?

  2. What are the differences in parental selection for F1 seeds between diploids and tetraploids? How might these selections impact the resulting hybrids?

Including discussions on these topics, along with schematic representations or detailed explanations, could enrich the reader's understanding and enhance the overall completeness of your research

Author Response

I have thoroughly reviewed your article and would like to commend you on the comprehensive nature of your work. The writing is clear, and the practical aspects are well-explained.

Response: Thank you for the commendation.

However, I have a few questions regarding hybrid seeds and heterosis in tetraploids and polyploids that I believe could enhance the depth of your manuscript. Specifically:

  1. How does the production of inbred lines differ between diploids and tetraploids? Are there any significant distinctions in this process?

Response: Thank you for this question. While discussing evidence for complementary gene action, the swift display of inbreeding depression in autotetraploids versus diploids is mentioned in section 4 Increasing multi-locus nuclear heterozygosity and heterosis. The original manuscript did not discuss this further and no mention of the practical aspects were addressed. Now, in the revised manuscript, there is added information in section 4 line 212 & 218 explaining that for some crops, breeding at the tetraploid level may not be possible due to unfavourable plant development (Riddle et al., 2006; Cohen et al., 2013). These references complement the already-present references to maize and rice autotetraploid breeding. In addition, line 223 now has an example of where inbreeding in an autotetraploid can be more difficult that in a diploid if the objective is to remove a single deleterious allele which is difficult to mask.

  1. What are the differences in parental selection for F1 seeds between diploids and tetraploids? How might these selections impact the resulting hybrids?

Response: As mentioned in response to (1), the updated manuscript now contains a discussion on potential barriers to choosing autotetraploid parent lines over diploid parent lines. There is a new schematic illustration of resulting selfed progeny from equivalent heterozygous autotetraploid and diploid lines at line 230.

Including discussions on these topics, along with schematic representations or detailed explanations, could enrich the reader's understanding and enhance the overall completeness of your research

Response: Thank you for this comment.

Reviewer 3 Report

Comments and Suggestions for Authors

The topic of heterosis remains relevant due to both its fundamental and applied significance. Moreover, many aspects of this issue are still far from being fully understood. I also appreciated the author’s approach, examining the problem from the perspective of nuclear heterozygosity.

To enhance the comprehensiveness of the review, I would recommend the author consider positive examples of the relationship between nuclear diversity and heterosis in diploid crops, of which there are many. For instance, a significant correlation was shown between genetic distances between parental lines and yield; however, for other traits under study, no such correlation was found (10.3844/ajabssp.2014.270.276). Similar results were previously obtained in sunflower (10.1007/s001220051366). Analogous cases have been recorded in barley (10.1111/j.1439-0523.2007.01367.x), carrots (10.1111/j.1439-0523.2011.01877.x), wheat, mustard, and many others. Earlier, about 28 studies demonstrated a correlation between heterosis and genetic distances between parents, which means heterozygosity in a hybrid (PMID: 15614727). All these cases cannot be ignored.

But why is there a correlation between heterozygosity and heterosis in diploids in some studies, but not in others? Here, I think it is also worth mentioning the theory that markers used to assess genetic distances and heterozygosity may be associated not only with loci that benefit hybrids but also with loci that have negative effects. There are a number of studies in this area that I would recommend the author to consider.

Author Response

The topic of heterosis remains relevant due to both its fundamental and applied significance. Moreover, many aspects of this issue are still far from being fully understood. I also appreciated the author’s approach, examining the problem from the perspective of nuclear heterozygosity.

Response: Thank you for the appreciation.

To enhance the comprehensiveness of the review, I would recommend the author consider positive examples of the relationship between nuclear diversity and heterosis in diploid crops, of which there are many. For instance, a significant correlation was shown between genetic distances between parental lines and yield; however, for other traits under study, no such correlation was found (10.3844/ajabssp.2014.270.276). Similar results were previously obtained in sunflower (10.1007/s001220051366). Analogous cases have been recorded in barley (10.1111/j.1439-0523.2007.01367.x), carrots (10.1111/j.1439-0523.2011.01877.x), wheat, mustard, and many others. Earlier, about 28 studies demonstrated a correlation between heterosis and genetic distances between parents, which means heterozygosity in a hybrid (PMID: 15614727). All these cases cannot be ignored.

Response: Thank you for the additional references, particularly PMID: 15614727, which has been included in section 1 Introduction line 51. It is encouraging to see that a number of the references that specifically analyse crop species with molecular markers have already been included in the original manuscript. Some new references have been added to the manuscript, namely those that quantify genetic distance accurately with molecular markers, such as those dealing with maize (Lee et al. (1989); Smith et al. (1990); Melchinger et al. (1990)) at lines 125-127. Some references are not relevant and hence are not included, e.g. when the analysis takes the “morphological distance” approach as opposed to nuclear heterozygosity, or, when preliminary analysis is further clarified by more detailed analysis at a later date, e.g. Zhang et al. (1994) investigate certain chromosomal regions in rice that could influence yield. They later perform a more detailed experiment as described in Zhang et al. (1995) and highlight that intercrossing specific groups is the best way of maximizing heterosis rather than a wide cross. It is in these instances where the 1995 reference is included while the 1994 reference is not, as the more up-to-date reference is important to the scope of the manuscript. In addition, some of the references deal with interspecies crosses (e.g. Eucalyptus) which is addressed briefly in the manuscript but is also beyond scope. Furthermore, some references are underscored by PMID: 15614727 as potentially favourable links between genetic distance and heterosis (e.g. Garcia et al. (1998)), yet the authors of this paper on melon specifically highlight that no heterosis measurements were taken. Again, in this instance it was determined not to include the reference in the manuscript.

The suggested references to sunflower are appreciated and have been added to the manuscript lines 139-142.

But why is there a correlation between heterozygosity and heterosis in diploids in some studies, but not in others? Here, I think it is also worth mentioning the theory that markers used to assess genetic distances and heterozygosity may be associated not only with loci that benefit hybrids but also with loci that have negative effects. There are a number of studies in this area that I would recommend the author to consider

Response: Thank you for the suggestion. This point fits nicely with the suggested reference above, PMID: 15614727. For a detailed overview of the use and misuse of molecular markers in hybrid development, I feel a separate paper to this one would be needed. However, section 5 Commercial breeding line 307 of the revised manuscript offers an explanation of why some diploid crops show heterosis upon wide crosses, and others have a neutral or negative experience.

Round 2

Reviewer 3 Report

Comments and Suggestions for Authors

The author made some corrections and responded in detail to the comments.